# Intra-Articular Injections in Knee Osteoarthritis: A Review of Literature

**DOI:** 10.3390/jfmk6010015

**Published:** 2021-02-03

**Authors:** Gianluca Testa, Serena Maria Chiara Giardina, Annalisa Culmone, Andrea Vescio, Matteo Turchetta, Salvatore Cannavò, Vito Pavone

**Affiliations:** Department of General Surgery and Medical Surgical Specialties, Section of Orthopaedics and Traumatology, P.O. “Policlinico Gaspare Rodolico”, University of Catania, 95123 Catania, Italy; gianpavel@hotmail.com (G.T.); serenamc.giardina@gmail.com (S.M.C.G.); annalisa.culmone@libero.it (A.C.); andreavescio88@gmail.com (A.V.); matteoturchetta1@gmail.com (M.T.); salvator.cannavo@gmail.com (S.C.)

**Keywords:** knee osteoarthritis, injection therapy, intra-articular injections, corticosteroids, hyaluronic acid, platelet-rich plasma

## Abstract

Knee osteoarthritis (OA) is a chronic, degenerative, and progressive disease of articular cartilage, producing discomfort and physical disability in older adults. Thirteen percent of elderly people complain of knee OA. Management options for knee OA could be divided into the following categories: conservative, pharmacological, procedural, and surgical. Joint replacement is the gold standard, reserved for severe grades of knee OA, due to its complications rate and increased risk of joint revision. A nonsurgical approach is the first choice in the adult population with cartilage damage and knee OA. Yearly, more than 10% of knee OA-affected patients undergo intra-articular injections of different drugs, especially within three months after OA diagnosis. Several molecules, such as corticosteroids injection, hyaluronic acid (HA), and platelet-rich plasma (PRP), are managed to reduce the symptoms of patients with knee OA. The aim of this review was to offer an overview of intra-articular injections used for the treatment of OA and report the conventional pharmacological products used.

## 1. Introduction

Osteoarthritis (OA) is one of the most common recurrent disabling joint disorders and represents a significant source of discomfort and disability in the Western world [1]. OA is a chronic, progressive, and degenerative disorder that involves the entire joint and presents bone and cartilage impairment that is characterized by variable inflammation and subchondral bone structural changes and damage of the protective articular cartilage [2]. More than 10% of the entire population is afflicted by this chronic joint condition, especially women between the ages of 50 and 60 years. It represents the main cause of disability in those > 65 years old, and the prevalence of arthritis and chronic joint symptoms increases with age [3]. There are two described OA forms: (1) idiopathic, or primary, which could be related to lifestyle factors or aging; and (2) secondary, which could be the consequence of several pathological conditions, for example, developmental and/or metabolic disorders, infection, or joint injury [4]. Every joint may be potentially affected by OA, although joints more frequently subjected to weight-bearing activities, such as knee joints (27%), are particularly affected. An increase in radiological exams of more than two percent each year is caused by knee OA in women over 55 years [5]. The etiology is not completely clear, but, in addition to age, other risk factors for OA include major trauma, joint overuse, and obesity, which could be largely involved in the progression of knee OA [4,6].

Glycosaminoglycans (GAGs), chondroitin sulfate, type II collagen, and hyaluronic acid (HA), inflammation, and lack of chondrocytes are part of a complex molecular interaction leading to the loss of structural components of cartilage and causing OA [7,8]. The progression of cartilage damage leads to exposure of the bone surface and to the formation of osteophytes and cartilage islands in the eburnated bony surface [9].

Diagnosis is largely clinical because radiographic findings do not always correlate with symptoms [10]. The main presentations are pain, tenderness, limitations in range of motion, joint effusion, and inflammation, which tend to present as disabilities in many patients [2]. The diagnosis of knee OA is typically based on the recognition of three symptoms (decreased function, stiffness, and constant knee pain) and three signs (limited movement, crepitus, and bony enlargement) [11]. The pain is usually correlated to activity, worsening with weight-bearing activities and improving with rest. Ultimately, the last stage is joint failure [4]. Radiographs are often used as the “gold standard” to confirm a diagnosis and to exclude other disorders. Knee x-rays in anteroposterior and lateral views show joint space narrowing, osteophyte formation, subchondral degenerative cysts, subchondral sclerosis, and joint destruction [12].

However, the correlation between radiographic OA severity and symptoms is low. In fact, many people with typical radiographic signs of OA have poor or negative symptoms [4].

OA treatment aims are to decrease pain, preserve mobility, and reduce disability, thereby improving function [4]. Management options could be divided into four typologies. Safe and less aggressive and less costly treatment should be considered for patients with OA before proceeding with invasive treatment [12]. In terms of non-pharmacological modalities that can result in short-term reduction of pain, physical therapy, such as exercises for muscle strengthening and range-of-motion exercises, biomechanical interventions, weight loss for overweight or obese patients, and therapeutic ultrasound [1,12] can be recommended. 

Pharmacological therapy consists of analgesics and anti-inflammatory agents that are administered gradually. According to guidelines, therapy should start with acetaminophen (up to 4 g/day), which is suggested as first-line systemic treatment for symptomatic OA, because its effectiveness has been found to be similar to nonsteroidal anti-inflammatory drugs (NSAIDs), but with lower gastrointestinal adverse effects and a stronger association with maintenance of warfarin [13,14]. NSAID therapy (ibuprofen, naproxen, diclofenac) is recommended when symptoms are moderate to severe or when there is no response to acetaminophen. NSAIDs are the most frequently utilized pharmacological molecules, although their use is limited because of their high incidence of gastrointestinal side effects, such as peptic ulcer and gastrointestinal bleeding, but also renal dysfunction and blood pressure elevation [12,15]. The role of cyclooxygenase-2 inhibitors (COX-2) is to reduce the gastrointestinal adverse effects and to improve the safety profile, but cyclooxygenase-2 inhibitors are associated with higher cost and cardiovascular complications, such as myocardial infarction and stroke [4,12,16]. 

Opioids (such as tramadol) may be used to treat OA pain in patients who do not respond to acetaminophen or NSAID therapy; it is also used to reduce the side effect of these drugs. Opioids should be provided starting at the lowest effective dose and carefully monitored to avoid dependence or abuse; however, they can cause chronic constipation and increase the risk of falls in older patients [12,17].

Intra-articular injection can primarily provide short-term relief from symptoms and lead to improvements in pain and function [12]. Several molecules are used to relieve knee OA symptoms, such as corticosteroid injection, hyaluronic acid (HA), and platelet-rich plasma (PRP). 

Pharmacological and non-pharmacological treatment could decrease the symptoms, especially pain, inflammation, and immobility, but the most successful treatment is total joint replacement (TJR). Total knee arthroplasty is an invasive procedure that is indicated in patients who present with severe persistent knee pain after six months of non-operative approach [18]. Nonsurgical treatment approaches play a central role in the elder population affected by cartilage damage and OA of the knee, due to the restricted TRJ lifespans and the joint revision risk [5]. 

The aim of this review was to offer an overview of the intra-articular injections used for the treatment of OA and report the conventional pharmacological products used.

## 2. Intra-Articular Injection of the Knee

Intra-articular injection of the knee may be a viable option for patients who do not tolerate pharmacological oral therapy, when drugs are no longer effective, or for those who want to delay or avoid surgical treatment. The injections should be as painless as possible. Proper injection site choice is mandatory, in fact, medial infrapatellar and suprapatellar were found to be more painful than lateral infrapatellar injection. In the medial subpatellar approach, the soft tissues are thinner and require traversing the vastus medialis obliquus. In the lateral patellofemoral approach, the patient lies supine with the joint in mild flexion. The needle should be inserted between the patella and femoral condyle until reaching the midpoint of the patellar equator [19]. 

In the case of severe patellofemoral compartment osteoarthritis or high body mass index subjects, a medial or lateral to the patellar tendon anterior approach is recommended. Planning the needle trajectory, identifying the target site, and avoiding adjacent neurovascular structures and tendons are mandatory, as is the proper recognition of different encountered tissue types during the needle injection. The right antiseptic precautions are very important for avoiding major complications during intra-articular injection of the knee [20]. 

Guidelines suggest that the ideal needle size should be a 22-gauge needle with a length between 1.5 and 3.5 in, and it is also recommended to change needles between drawing the medication and injecting it into the joint [19,20]. The most frequently reported complications are pain and bleeding at the injection site, while septic arthritis is a rare major complication of intra-articular steroid injections. Bacterial arthritis, generally involving *Staphylococcus aureus*, was observed to occur in <1 in 10,000 cases in one series and is associated with a >15% mortality and residual impairment of joint function in 50% of the survivors. The main risk factors are joint surgery, rheumatoid arthritis and other joint diseases, diabetes mellitus, skin defects or infections, advanced age, and immunosuppressive medication. Elective injections should be deferred if the patient would be subject to bacteremia. Knowledge of the sepsis risk is important not only for taking the right precautions in performing intra-articular injections of the knee but also for medico-legal reasons [19,20].

### 2.1. Corticosteroids

Intra-articular steroid injection (IASI) of the knee is effective for short- to medium-term treatment of joint pain [19]. Through the downregulation of aggrecans and collagenases, agents/modulators of proinflammatory mediators and mononuclear cells, corticosteroids reduce against synovial inflammatory phase [21]. The mechanism of action is complex and leads to a decrease in synovial blood flow and the number of leukocytes, as well as the release of inflammatory mediator [19]. Joint inflammation in knee OA is associated with progression of cartilage damage, therefore IASI might reduce disease progression [21]. 

There is a large variety of corticosteroids on the market, such as triamcinolone acetonide (Kenalog), dexamethasone (Decadron) LA, betamethasone (Celestone), and methylprednisolone acetate (Depo-Medrol). The most frequently used are methylprednisolone acetate (DepoMedrol) and triamcinolone acetonide (Kenalog). Their typical dosage is 40 mg, with an interval of at least three months between injections, as shown in Table 1 [22].

Because the clinical duration of an effect is inversely related to solubility, the most commonly applied steroid is an insoluble one (triamcinolone acetonide), while more soluble steroids, such as dexamethasone, are preferable for superficial injection because they are less likely to cause subcutaneous fat atrophy or depigmentation of skin [19]. IASI could be administered with an anesthetic (lidocaine and ropivacaine are usually used, as they provide more rapid onset and longer-lasting anesthetic properties) to increase the injectate volume accordingly in order to distribute the steroid throughout the joint in addition to reducing patient discomfort and providing some immediate relief [19,23].

According to many studies, IASI provides short-term relief for at least one to eight weeks, and the usual practice is limited to four injections annually [12].

A potential side effect after anesthetic injection is the flare-up of symptoms within the first 24 h, which is generally followed by an improvement over baseline by 48 h [12]. Furthermore, recent studies have shown how lidocaine or bupivacaine hydrochloride can adversely affect a joint via chondrotoxicity [22] and significant cartilage volume loss with no significant difference in long-term knee pain [21]. Comparing triamcinolone to saline solution, McAlindon et al. founded a greater rate of cartilage loss and thickness in the triamcinolone group. At the same time, no significant differences in subchondral tibia, hip, or bone mineral density were noted between the two groups [21]. According to a radiological study, documented adverse joint events after IASI include accelerated osteoarthritis progression, subchondral insufficiency fracture, complications of osteonecrosis, and rapid joint destruction with bone loss [24].

### 2.2. Hyaluronic Acid (HA)

In 1997, the Food and Drug Administration approved viscosupplementation as a conservative technique in OA management, and in 2000, the American College of Rheumatology (ACR) introduced viscosupplementation as a therapeutic choice by a guideline for pain management in knee OA [25]. HA is natural high-molecular-weight glycosaminoglycan contained in the synovial fluid and extracellular matrix that is formed from chains of repeating disaccharide units. Its function is to lubricate the joint and absorb shocks during movements, thus increasing synovial fluid viscosity. HA also contributes to the inhibition of nociceptors and enzymatic cartilage degradation. Furthermore, exogenous HA could stimulate the synthesis of endogenous HA, and it may also act as a free-radical scavenging agent [26]. Molecular fragmentation, abnormal synoviocyte production, and synovial fluid dilution secondary to effusion are the reasons for decreasing HA concentration and molecular weight. However, the mechanical influence of exogenous HA cannot account for prolonged benefits, noted in experimental studies; in fact, the intra-articular hyaluronic acid (IAHA) preparation influences the HA clearance within a few days [27].

There are several available varieties of HA with different molecular weights, which are classified into three types: (1) low (500–730 kDa); (2) intermediate (800–2000 kDa); and (3) high (2000–6000 kDa), including crosslinked formulations of HA (Table 2). 

The increased molecular weight reduces the velocity of enzymatic degradation and improves the permanence in the joint, as well as the molecular configuration. Higher molecular weight (HMW) HA has been reported to provide greater anti-inflammatory and proteoglycan synthesis effects, as well as joint lubrication and viscoelasticity maintenance. Another study demonstrated the favorable matrix metalloproteinase (MMP) inhibitory effect of lower molecular weight (LMW) products [28].

Various randomized controlled trials have assessed the efficacy of HA for pain and joint function. The benefits of HA depend on time, with the maximal effect on pain at 8 to 24 weeks from the time of injection. Studies demonstrated that HA for knee OA was of greater benefit in those with less severe radiographic changes with the second course of HA [29].

Non-animal stabilized hyaluronic acid (NASHA) has been shown to be effective in relieving joint pain in preclinical animal models and has been demonstrated to have a half-life of 32 days in rabbit knee joints [29]. Moreover, according to human clinical trials, significant reductions from baseline pain, as well as improvements in physical function and joint stiffness from baseline levels, were observed at 26 weeks after a single injection of NASHA [30].

IAHA is suggested in refractory patients after continuous or intermittent treatment with as acetaminophen, NSAIDs, and symptomatic slow-acting drugs [31]. IAHA has been recognized as a reliable and secure therapy approach for knee OA. Previous studies suggest that the adverse events, such as pain or swelling, associated with IAHA nearly always occur at the site of injection or within the joint and are just as likely to occur in the placebo-treated patients as in the actively treated individuals. Serious adverse events are rare. The Food and Drug Administration premarket approval database revealed no post-marketing reports concerning unexpected adverse events [32].

### 2.3. Platelet-Rich Plasma (PRP)

PRP is the result of the preparation of autologous plasma enriched with a platelet concentration [33]. PRP supplies and releases cytokines, growth factors, and α-granules, which can offer a recovering stimulus and promote healing and tissue repair [34]. The PRP injection can promote the release of fibrinogen, interleukin-1 receptor antagonist (IL-1RA), tissue growth factors (TGFs), platelet-derived growth factors (PDGFs), and vascular endothelial growth factors (VEGFs) [35]. These growth factors have local and systemic involvement, encouraging the inhibition of catabolic enzymes and cytokines, modulating inflammation and local angiogenesis, and recruiting local stem cells and fibroblasts to sites of damage, and inducing healthy nearby cells to manufacture greater numbers of growth factors [36]. 

In knee OA, PRP was proven to halt chondrocytes catabolic activity, which is important for the reduction of the chondrocyte apoptosis rate, also resulting in a decrease in the loss of the cartilage matrix secreted by cartilage cells and an increase in cartilage height [34]. PRP is an autologous mixture of highly concentrated platelets and associated growth factors and other bioactive components produced by centrifugal separation of whole blood. The growth factors released by PRP promote cell recruitment, proliferation, and angiogenesis, with consequent decreased release of critical regulators of the inflammatory process and a reduction of inflammatory enzymes. This could induce a regenerative response, improving the metabolic functions of damaged structures and promoting a positive effect on chondrogenesis and mesenchymal stem cell proliferation [35,36,37]. Despite the suggestion by some authors of using PRP as the first choice in intra-articular injections [37], the indications of PRP injection for knee OA are controversial, and there is no common consensus among orthopedic surgeons for it use. 

Numerous PRP preparation techniques, platelet counts, numbers of injections, and uses of anticoagulants and activating agents, in addition to patient sex, personal physical characteristics, and severities of OA, have been reported in literature [38]. The duration of the beneficial effects of PRP injections are unclear, and current evidence indicates that for at least 12 months PRP can improve pain relief and functional improvement in patients with symptomatic knee OA, but some authors have described good score values up until 24 months from the beginning of the treatment [39]. Compared to other injective therapies (HA, IASI, and saline), treatment with PRP was found to be clinically superior in reducing OA-related pain symptomatology and increasing the functional outcomes with similar or less risks of adverse events. A recent meta-analysis found that platelet concentrates offer a benefit that, although not significant at the earlier follow-ups, exceeds over time both the placebo effect and the improvement offered by other intra--articular options at 12 months, without an increased risk of adverse events [40]. No previous authors have documented major adverse events, and minor and transient events reported consist of local pain, headache, gastritis, stiffness, bleeding, syncope, tenderness, dizziness, nausea, tachycardia, and injection site bruising and/or swelling [41].

The lack of standardization and the additional research that is needed to examine how the inclusion of leukocytes, activation, and platelet concentration affect therapeutic efficacy [42] beyond the cost-effectiveness of PRP [43] are currently the greatest limiting factors for therapy.

### 2.4. Future Direction

Mesenchymal stem cells (MSCs) are harvested from adipose tissue or marrow bone and they produce growth factors such as TGF, VEGF, and FGF, allowing the differentiation of various cell types with the potential to differentiate into chondral tissue and contributing to tissue repair [44,45]. Adipose-derived stromal cells (ADSCs) share many features with bone marrow stem cells, such as paracrine activity [46] and strong angiogenic activity [47,48]. Moreover, ADSCs have been noted to have high chondrogenic differentiation features [49]. According to the few published articles, OA treatment with ADSC is safe and effective for disease stabilization and pain relief when used with multiple infiltrations or with high dosage [50,51]. The most effective dosage seems to be greater than 5 *×* 10^7^/mL, and the time needed to show an improvement in the symptoms and cartilage volume varies from 24 to 72 weeks [51]. On the other hand, some trials described the specific facilities of hospital laboratories needed to isolate and expand ADSCs; others reported on the necessity of cryopreservation, with no clear cost estimations [49,50,51].

## 3. Conclusions

Knee OA is the most common disabling joint disease and represents a significant social burden. According to the severity of the pathology, intra-articular injections of corticosteroid, HA, and PRP represent optimal treatment options with minor adverse events for pain relief and symptom alleviation and can delay surgical treatment. However, the choice of timing, severity, molecules, and the efficacy time are the current major limitations of the treatment. Further studies are needed to investigate new molecules and/or associated therapies in knee OA intra-articular injection treatment.

## Figures and Tables

**Table 1 jfmk-06-00015-t001:** Varieties of corticosteroids and dosage.

Product	Dosage	Relative Anti-Inflammatory Potency	Approximate Duration of Action (Biological Half-Life in Hours)
Methylprednisolone acetate	20–80 mg	5	12–35
Triamcinolone acetonide	10–40 mg	5	12–35
Triamcinolone hexacetonide	10–20 mg	5	12–35
Dexamethasone acetate	8 mg	25	36–72
Dexamethasone sodium	8 mg	25	36–72
Hydrocortisone acetate	10–25 mg	1	8–12
Betamethasone sodium phosphate and acetate	0.25–2 mL	20	36–72

**Table 2 jfmk-06-00015-t002:** Varieties in the molecular weight and dosage of hyaluronic acid (HA).

Brand Name Manufacturer	Generic Content	Molecular Weight (kDa)	Dosage
Hyalgan (Fidia Pharma)	1% sodium hyaluronate	500–730	20 mg weekly (five injections)
Synvisc (Sanofi)	0.8% hylan G-F 20	6000	16 mg weekly (three injections)
Synvisc-One (Sanofi)	0.8% hylan G-F 20	6000	48 mg one-time injection
Supartz (Bioventus)	1% sodium hyaluronate	620–1170	10 mg weekly (five injections)
Euflexxa (Ferring B.V.)	1% sodium hyaluronate	2400–3600	20 mg weekly (three injections)
Gel-One (Zimmer)	1% cross-linked hyaluronate	Not disclosed	30 mg one-time injection
Orthovisc (DePuy Synthes)	1.5% sodium hyaluronate	1000–2900	30 mg weekly (three to four injections)
Monovisc (Anika/Pendopharm)	2.2% cross-linked hyaluronan (proprietary cross-linking agent)	1000–2900	88 mg one-time injection
GenVisc 850 (Adant) (OrthogenRx)	1% sodium hyaluronate	620–1170	25 mg weekly (five injections)
Hymovis (Fidia Pharma)	0.8% hexadecylamide derivative of hyaluronan	500–730	24 mg weekly (two injections)
Gelsyn-3 (Gel-Syn) (Bioventus LLC)	0.84% sodium hyaluronate	1100	16.8 mg weekly (three injections)
Durolane (Bioventus LLC)	non-animal stabilized hyaluronic acid	Not disclosed	60 mg one-time injection

## Data Availability

No new data were created or analyzed in this study. Data sharing is not applicable to this article.

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
