# Peer review of "Intra-Articular Injections in Knee Osteoarthritis: A Review of Literature"

_jfmk, 2021, doi:10.3390/jfmk6010015_

Round 1

Reviewer 1 Report

This is a brief and not very comprehensive review of injectable treatments of knee osteoarthritis. There are considerable differences in ease of delivery and cost for each of the described techniques which is not addressed

Author Response

This is a brief and not very comprehensive review of injectable treatments of knee osteoarthritis. There are considerable differences in ease of delivery and cost for each of the described techniques which is not addressed

Thank for your manuscript revision. Several changes were included, and more recent findings were reported in the study, hoping that you will appreciate.

Reviewer 2 Report

The aim of the review manuscript by Testa et al was to analyze and report the conventional pharmacological products used with intra-articular injections for osteoarthritis (OA).

The review starts with a global overview of OA, how to set the diagnosis, shorty mentioning all treatment options, followed by a more specific description of intra-articular injections with more details on corticosteroids, hyaluronic acid and PRP injections. The structure is logical, clear and easy to grasp.

However, there are points that really need to be addressed:

  1. This reviewers major concern is the lack of novelty and this review brings nothing new. Quite recently a very thorough meta-analysis has been performed on PRP injections (compared to corticosteroids and hyaluronic acid) and it describes basically the same, but then with a better quality analysis added (Filardo et al., Cartilage. 2020; doi:10.1177/1947603520931170). The novelty could be improved if the review paper is given a more specific goal than just reporting conventional pharmaclogical products; the aim of analyzing is not even met. One option could be to add how these pharmacological products act on cells for instance. However, this is just an option and the author´s should think about it.
  2. Especially the abstract should be checked by a (native) English speaker
  3. Abstract, line 12, what is meant with procedural typology of treatment?
  4. Line 41; collagen type II should be type II collagen
  5. Line 41 - 43; this is too vague. All these points are involved in OA, but it should be made more clear what is there role in OA.
  6. Lines 157 - 160: this is major discussion point, there are serious concerns about using corticosteroids injections and should get more attention. There are more (also original research) papers out there showing that it can be really detrimental for cartilage; McAlindon et al., JAMA. 2017;317(19):1967-1975 and Kompel et al., Radiology. 2019;293(3):656-663
  7. Table 2; the list is not complete. This should be improved if brand and manufacturers names are provided (think of Durolane for instance). Also, the table draws the attention to the various molecular weights of HA, but this is not discussed anywhere, just briefly mentioned.

Author Response

The aim of the review manuscript by Testa et al was to analyze and report the conventional pharmacological products used with intra-articular injections for osteoarthritis (OA).

The review starts with a global overview of OA, how to set the diagnosis, shorty mentioning all treatment options, followed by a more specific description of intra-articular injections with more details on corticosteroids, hyaluronic acid and PRP injections. The structure is logical, clear and easy to grasp.

However, there are points that really need to be addressed:

Q1) This reviewers major concern is the lack of novelty and this review brings nothing new. Quite recently a very thorough meta-analysis has been performed on PRP injections (compared to corticosteroids and hyaluronic acid) and it describes basically the same, but then with a better quality analysis added (Filardo et al., Cartilage. 2020; doi:10.1177/1947603520931170).

A1) thanks for your suggestion. The reference was included in the text and the issues discussed.

Q2) The novelty could be improved if the review paper is given a more specific goal than just reporting conventional pharmaclogical products; the aim of analyzing is not even met. One option could be to add how these pharmacological products act on cells for instance. However, this is just an option and the author´s should think about it.

A2) thanks for your suggestion. A new section “future direction” was added.

Q3) Especially the abstract should be checked by a (native) English speaker

A3) Following your suggestion, an extensive English editing was made.

Q4) Abstract, line 12, what is meant with procedural typology of treatment?

A4) thanks for your suggestion. the phrase was re-written.

Q5) Line 41; collagen type II should be type II collagen

A5) thanks for your suggestion. the typo was corrected.

Q6) Line 41 - 43; this is too vague. All these points are involved in OA, but it should be made more clear what is there role in OA.

A6) Thanks for your suggestion. The points were clarified.

Q7) Lines 157 - 160: this is major discussion point, there are serious concerns about using corticosteroids injections and should get more attention. There are more (also original research) papers out there showing that it can be really detrimental for cartilage; McAlindon et al., JAMA. 2017;317(19):1967-1975 and Kompel et al., Radiology. 2019;293(3):656-663

A7) thanks for your suggestion. The references wer included in the text and the issues discussed.

Q8) Table 2; the list is not complete. This should be improved if brand and manufacturers names are provided (think of Durolane for instance). Also, the table draws the attention to the various molecular weights of HA, but this is not discussed anywhere, just briefly mentioned.

A8) thanks for your suggestion. The discussion was amplified and new molecules added

Round 2

Reviewer 1 Report

The new section at the end on future direction is a little misguided. The last sentence suggests that to acheive adequate cell concentration a cell expansion laboratory is always required. There are several products that have been available for a long time (MarrowStim - Biomet) that concentrate cells by various techniques (eg centrifuge) so that a one stage procedure can be performed without the complexities and cost of cellular expansion. Exciting newer technologies (eg Cell Select machines) may be transfereable to the operating theatre soon...

Author Response

Thanks for your suggestion. The phrase was partially re-written.

Reviewer 2 Report

To this reviewer´s opinion, the manuscript has improved with these (relatively minor) changes. Although not meant to put it in this way, the subchapter on future direction does provide more depth and make the story complete.

However, to this reviewer´s opinion, there is one point left that needs to be addressed. The aim, as described in lines 98-99, was "to analyze and report the conventional pharmacological products used with intra-articular injections for OA". However, no analyses have been performed, it is just that an overview is provided. Therefore, just to be correct and provide readers a good expectation, I would suggest to change the aim in both the abstract and introduction. Something as "the aim was to provide an overview and report the conventional pharmacological products used with intra-articular injections for OA" would be sufficient (your manuscript, so please write it in your own words).

Author Response

Thanks for your suggestion. The requested modifies were made